# Thermal-intensified interfacial polymerization enables ultra-selective reverse osmosis membrane for toxic micropollutant removal

Shenghua Zhou [1], Lu Elfa Peng[1], Wenyu Liu [1], Hao Guo [2,3] ✉ & Chuyang Y. Tang [1] ✉

Polyamide reverse osmosis (RO) membranes are widely used in seawater desalination and wastewater reuse, yet often fail to remove small toxic micropollutants. Herein, we develop a thermal-intensified interfacial polymerization (TIP) strategy to fabricate highly selective RO membranes against various micropollutants. Facile heating accelerates amine monomer diffusion, intensifying interfacial polymerization to form a highly crosslinked polyamide membrane. The resultant TIP membrane achieves rejection of 90.8%, 98.0%, and > 99% for boron, arsenite, organic micropollutants at neutral pH, respectively. Meanwhile, high temperature facilitates interfacial degassing and promotes the formation of more extensive nanovoids within the polyamide. These nanovoids increase membrane surface area and optimize water transport pathways, thereby boosting water permeance. The combination of high solute rejection and water permeance enables the membrane to achieve high water-micropollutant selectivity (e.g., water-boron). Our study demonstrates that TIP technique holds a great promise to fabricate ultra-selective polyamide membranes for desalination and wastewater reuse.

Sustainable clean water supply is vital for public health and socioeconomic development[1,2], yet nearly one third of the global population lacks reliable access to safe drinking water[3]. Reverse osmosis (RO) has been widely used to recover clean water from unconventional water sources (e.g., seawater, brackish groundwater, and wastewater) through desalination and water reuse[4–6]. Today, RO produces over 88 million cubic meters of desalted water per day[7,8], showing its immense potential to augment clean water supply. Existing thin film composite (TFC) polyamide RO membranes can achieve reasonable water permeance and > 99.0% NaCl rejection[9,10]. Nevertheless, they often showed poor rejection of some toxic and harmful micropollutants with severe environmental and health concerns[11,12]. For example, boron, a small neutral compound (molecular weight = 61.8 g mol⁻¹ and pKa = 9.2), is ubiquitously found in seawater. Current commercial RO membranes often show insufficient boron rejection of <80% at circumneutral pH[12–16], which needs a second-pass RO polishing step to meet the boron concentration requirement for drinking and agricultural uses[17,18]. Likewise, inadequate rejection of endocrine disrupting compounds (EDCs) and antibiotics raises critical concerns in water reuse[11,19], while unsatisfactory rejection of arsenite (As (III)) could be problematic for groundwater treatment[20,21]. To address these challenges, RO membranes with high selectivity against micropollutants are urgently needed.

[1]Department of Civil Engineering, The University of Hong Kong, Pokfulam, Hong Kong SAR, China. [2]Institute of Environment and Ecology, Shenzhen International Graduate School, Tsinghua University, Shenzhen, China. [3]Guangdong Provincial Key Laboratory of Carbon Fixation and Sinks, Department of Education of Guangdong Province, Shenzhen International Graduate School, Tsinghua University, Shenzhen, China. ✉e-mail: guohao@sz.tsinghua.edu.cn; tangc@hku.hk

The selectivity of a TFC RO membrane strongly depends on the crosslinking degree (and thus the effective pore size) of its polyamide rejection layer, which is fabricated by interfacial polymerization (IP) between *m*-phenylenediamine (MPD) and trimesoyl chloride (TMC)[22–24]. Numerous efforts have been made to narrow the effective pore size of the polyamide layer to improve membrane selectivity, such as incorporating additives into monomer solutions and post-treating the fabricated membranes[25–27]. However, selectivity enhancement is often at the expense of sacrificed water permeance, commonly known as the trade-off effect between permeance and selectivity[9,10,28]. The key to overcome this effect is to tailor the chemistry and nanostructure of polyamide during its formation through IP reaction. Raising the reaction temperature can significantly accelerate the diffusion of amine monomers from the aqueous phase to the organic phase, thus promoting the IP process[29]. Meanwhile, molecular motion would become more rapid at higher temperature due to the increased kinetic energy, which leads to more frequent collisions between the reactant molecules. These effects could significantly promote the IP reaction to form a more crosslinked polyamide thereby potentially improving membrane selectivity. Therefore, we hypothesize that increasing temperature could facilitate the IP reaction to form a highly selective polyamide membrane.

Herein, we synthesized an ultra-selective polyamide RO membrane for toxic micropollutant removal via a thermal-intensified IP (TIP) method. By heating the organic solvent of Isopar G (boiling point = 166 °C) to 100 °C, the resultant polyamide membrane showed enhanced crosslinking degree and increased nanovoid fraction compared with that fabricated without heating. Consequently, the TIP membrane exhibited high rejection of various micropollutants (e.g., 90.8% for boron, 98.0% for As (III), and >99.0% for EDCs and antibiotics) at neutral pH, which exceeds the reported rejection values in the literature. At the same time, it showed significantly enhanced water permeance thanks to the larger filtration area and optimized water transport pathway induced by the extensive interior nanovoids within polyamide. These features enable the membrane to achieve ultra-selective removal of micropollutants. Our findings indicate that the TIP is an effective strategy to fabricate ultra-selective RO membranes toward membrane-based desalination and water reuse.

## Results

### Separation performance of TIP membranes

The TIP RO membranes were fabricated by performing the IP reaction between MPD (dissolved in water at room temperature of ~25 °C) and TMC (dissolved in Isopar G at different temperatures). The formed membranes were named as TIP0, TIP25, TIP50, and TIP100 corresponding to the temperature of Isopar G (i.e., 0 °C, 25 °C, 50 °C, and 100 °C, respectively). As shown in Fig. 1, TIP membranes fabricated at higher organic solvent temperature showed greatly improved separation performance. With an increase in temperature from 0 °C to 100 °C, water permeance was almost tripled (Fig. 1A), and a simultaneous enhancement in NaCl rejection was also observed (Fig. 1B). Specifically, TIP100 exhibited an attractive combination of water permeance (1.8 L m$^{-2}$ h$^{-1}$ bar$^{-1}$) and NaCl rejection (99.1%). It is worthwhile to note that membrane rejection can be further enhanced by post-heating (e.g., soaking nascently formed membranes in a 50 °C water bath for 10 min in this study) to facilitate the secondary crosslinking of polyamide chains[30]. In this regard, we prepared two post-treated polyamide membranes (TIP25-W and TIP100-W, with W indicating heat treatment in water), and both membranes showed higher NaCl rejection compared to their respective counterparts without post-heating (Fig. 1B). Whereas TIP25-W exhibited comparable water permeance with TIP25, TIP100-W showed a 17.6% reduction in water permeance compared with TIP100 (Fig. 1A). For TIP25-W that is post-treated at relatively low temperature, its crosslinking degree remained relatively low (Supplementary Fig. 10). In contrast, TIP100-W reached a very high

crosslinking degree of 89.1% after the heat treatment, which partially explains moderate loss of water permeance. Furthermore, the surface of TIP100-W became significantly more hydrophobic compared to TIP100 (Supplementary Fig. 9), which may further increase the resistance to water transport. Moreover, increasing organic solvent temperature and additional heat treatment also improved the rejection of As (III) and boron, respectively. We further evaluate membrane rejection to a variety of organic micropollutants with critical concerns for public health such as EDCs and antibiotics (Fig. 1C). Both TIP100 and TIP100-W achieved remarkable rejection of these compounds. Even for methylparaben, a neutral compound with molecular weight as small as 152.2 g mol$^{-1}$, its rejection by TIP100 and TIP100-W reached 98.9% and 99.1%, respectively. Notably, TIP100-W achieved high rejection for a broad spectrum of contaminants, e.g., 99.5% for NaCl, 90.8% for boron, 98.0% for As (III), and >99.0% for all the nine organic micropollutants at pH7, consistently overperforming against other membranes reported in the literature (Fig. 1D). These results demonstrate the benefit of TIP technique for enhanced membrane rejection of toxic contaminants thereby ensuring the safety of product water.

### Thermal effects on the formation of TIP membranes

As shown in Fig. 2A, the ultraviolet-visible (UV) absorbance of MPD monomers in 100 °C Isopar G was approximately three times higher than that in 0 °C, implying accelerated MPD monomer diffusion at higher temperature. More MPD supply can facilitate the IP reaction to form a more crosslinked polyamide with crosslinking degree increasing from 17.7% at 0 °C to 78.4% at 100 °C (Fig. 2B). Consistently, the density of ionized carboxyl groups (formed by hydrolysis of unreacted acyl chloride groups) decreased from 38.2 nm$^{-2}$ for TIP0 to 12.5 nm$^{-2}$ for TIP100 (Fig. 2C), resulting in a less negatively charged membrane surface (Fig. 2D). It is worthwhile noting that these carboxyl group densities were estimated based on the projected membrane area, as commonly practiced in the literature[31,32]. However, since the actual surface area of the polyamide layer increases with higher Isopar G temperature, the carboxyl group density normalized by the actual surface area is even lower (28.1 nm$^{-2}$ for TIP0 vs. 7.2 nm$^{-2}$ for TIP100, Supplementary Fig. 8). In addition, post-heating further enhanced membrane crosslinking degree (Supplementary Fig. 10) due to the secondary crosslinking of polyamide[30]. To further investigate the effects of temperature on the pore size of polyamide membranes, Doppler Broadening Energy Spectroscopy (DBES) characterizations was performed to determine the S parameter of TIP25 and TIP100 membranes. The TIP100 membrane has a lower S parameter than that of TIP25 membrane (Fig. 2E). According to the literature[33,34], a lower S parameter suggests a smaller pore size thereby improving size exclusion effect of the membrane, which is consistent with the higher rejection of four neutral solutes by the TIP100 membrane (Fig. 2F). Consequently, the enhanced crosslinking degree (Fig. 2B) and reduced pore size (Supplementary Fig. 11) improved membrane rejection of various toxic micropollutants (Fig. 1B, C).

Increasing the temperature from 0 °C to 100 °C led to the enlargement of "leaf-like" features on membrane surface (Fig. 3A). Consistently, membrane surface roughness substantially increased from 45.2 nm for TIP0 to 88.9 nm for TIP100 as determined by AFM (Fig. 3B). In addition, transmission electron microscope (TEM) characterizations revealed the presence of more prominent nanovoids within the polyamide rejection layer formed at higher temperature (Fig. 3C). These nanovoids, formed by the encapsulation of degassed nanobubbles by the nascent polyamide during IP reaction[35,36], are enclosed between the polyamide film and the substrate (as illustrated by the red-highlighted area, Supplementary Fig. 15B). Notably, TIP100 membrane had the largest nanovoid size up to ~0.5 μm (Fig. 3C4), corresponding to its most extensive "leaf-like" features among the four TIP membranes (Fig. 3A). These enlarged nanovoids were formed during the intensified IP reaction at higher temperature. Briefly, the increased temperature

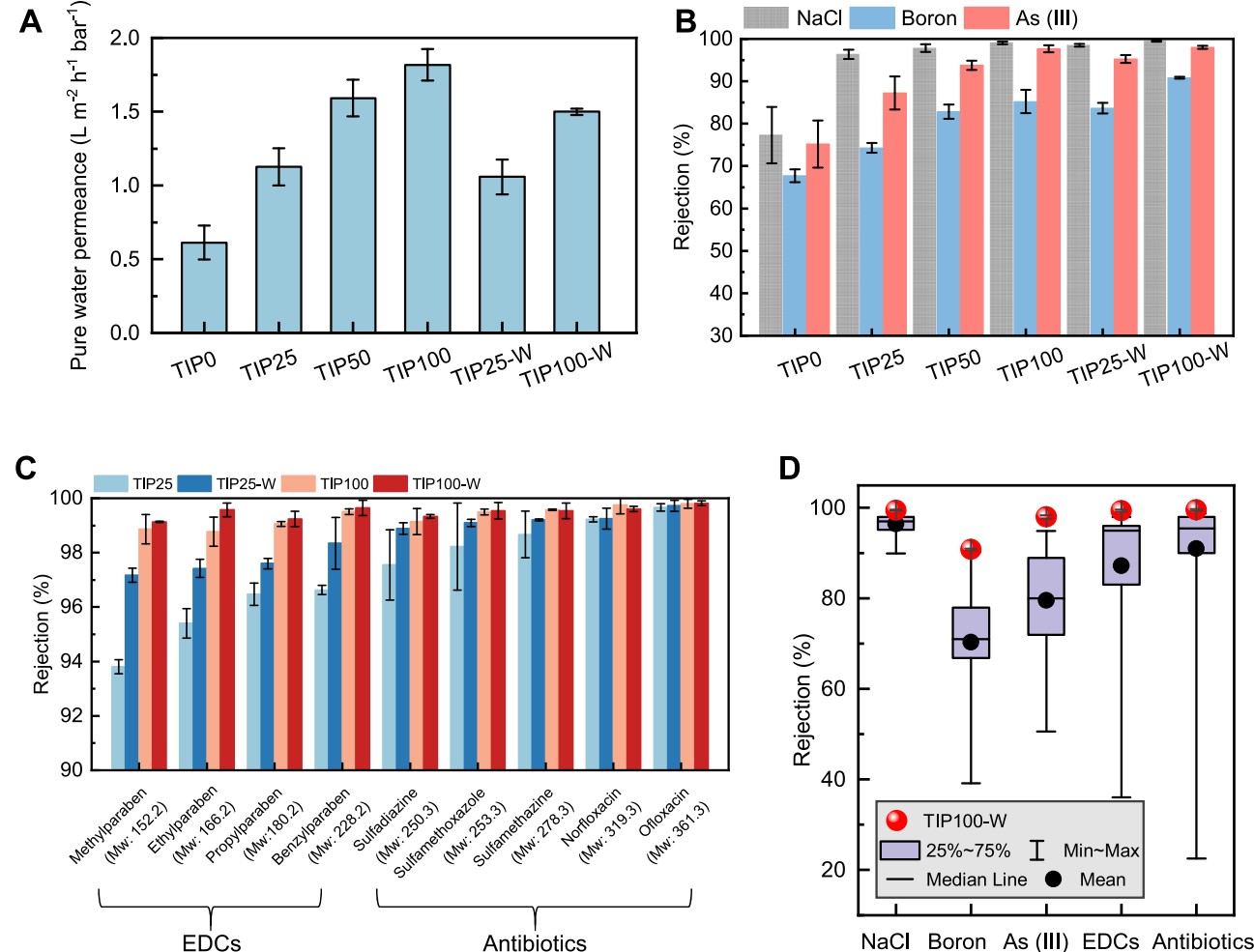

**Fig. 1 | Separation performance of various TIP RO membranes. A** Pure water permeance. **B** Rejection of NaCl, boron, and As (III). **C** Rejection of EDCs and antibiotics. **D** Comparison of NaCl, boron, As (III), EDC, and antibiotic rejection by TIP100-W with the literature data (Supplementary Tables 4–7). All the rejection data was acquired from the filtration tests at the neutral pH of 6-8. The error bars represent the standard deviation of the results obtained from at least three independent measurements of different membranes.

and H[+] (a byproduct of IP) could facilitate interfacial degassing of $CO_2$ nanobubbles from the aqueous solution ($HCO_3^- + H^+ \rightarrow^\Delta CO_2 \uparrow + H_2O$), which were further captured by the nascent polyamide thereby forming the extensive nanovoid structure[35–40]. The additional gas generated after the formation of this nascent polyamide film will have to escape from the back side, forming the back opening structures of the polyamide (Supplementary Fig. 13 and Supplementary Note 12)[35,40]. These back-side openings became larger at higher temperature conditions (e.g., 14.3 nm for TIP0 vs. 30.0 nm for TIP100, Fig. 3D, E). According to literature[39–41], the size of the openings is well correlated to the intensity of degassing. Furthermore, the back-side openings connect the nanovoids in the polyamide to the substrate pores (Supplementary Fig. 13)[35,39,40,42]. Meanwhile, the intensive release of nanobubbles also led to larger back opening (Fig. 3D, E, measurement of back pore size was shown in Supplementary Note 12) as a result of substrate-confined degassing[35]. Consequently, the TIP100 membrane exhibited a nanovoid fraction of 50.0%, which was over one order of magnitude higher than that of 4.2% for TIP0 (Fig. 3F, calculation of nanovoid fraction was shown in Supplementary Note 13). In addition, membrane surface area ratio was also significantly increased at higher temperatures (Fig. 3G), which could provide greater effective filtration area. Such increased nanovoid fraction and surface area could optimize water transport pathway with reduced resistance[43], leading to enhanced membrane water permeance (Fig. 1A).

## Selectivity and fouling behavior of TIP membranes

A high organic phase temperature greatly enhanced membrane selectivity. Specifically, the TIP100 membrane demonstrated higher water-boron selectivity (Fig. 4A) than the reported data in existing literature. At the same time, this membrane exhibited high water-As (III) selectivity, which is better than most membranes reported in the literature (Fig. 4B). In comparison, membranes fabricated at lower temperature (i.e., TIP0, TIP25, and TIP50) showed relatively lower selectivity falling in the trade-off region. This result demonstrates the importance of high temperature on enhancing water-solute selectivity of polyamide RO membranes. Moreover, high selectivity not only improves the quality of product water but also eliminates the requirements for additional treatment (e.g., a second pass RO step for boron removal)[17,18]. It is also worth noting that post-treatment of the resultant membrane in a 50 °C water bath (i.e., TIP100-W) further enhanced its selectivity.

We further evaluated the fouling behavior of TIP25 and TIP100 membranes using humic acid (HA) as the foulant. TIP100 membrane demonstrated high antifouling performance (Fig. 4C) and less HA accumulation on its surface (Fig. 4D), despite its greater roughness (Fig. 3B) and hydrophobicity (Supplementary Fig. 9). The reduced fouling propensity can be attributed to the extensive nanovoid structures of the membrane[44–46]. Such structures could allow more uniform water transport and flux distribution near the extensive

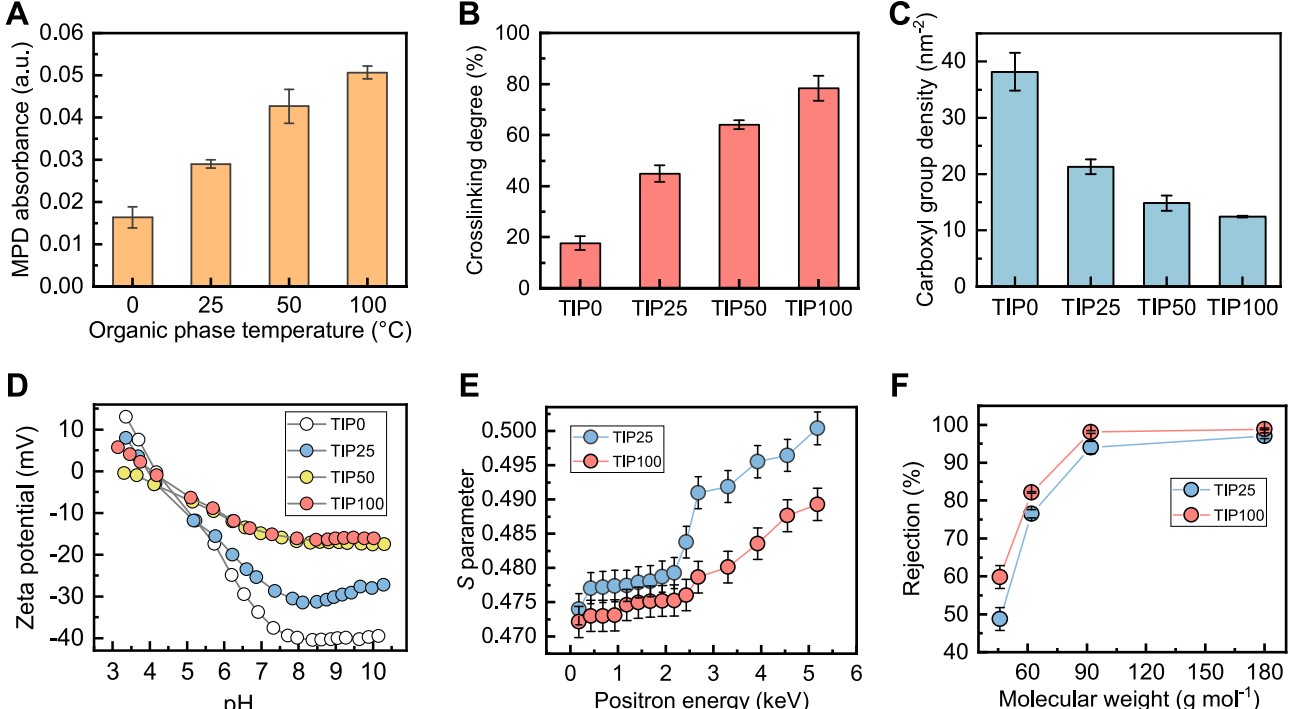

**Fig. 2 | Thermal effects on MPD diffusion and membrane properties. A** UV absorbance of MPD diffused in the Isopar G at different temperatures (i.e., 0 °C, 25 °C, 50 °C, and 100 °C). The MPD diffusion measurement was not taken from the actual IP reaction solutions but in a separate control test to evaluate MPD diffusion across the water-Isopar G interface (Supplementary Fig. 4). **B** Crosslinking degree of the formed polyamide TIP membranes (i.e., TIP0, TIP25, TIP50, and TIP100, respectively). The crosslinking degree ($n$) can be calculated based on O/N ratio ($y$) obtained from X-ray photoelectron spectroscopy (XPS) characterization following

by $n = (4 − 2y)/(1 + y)$[62]. **C** Ionized carboxyl group density of TIP membranes. The density was determined using a reported silver binding method[31,51]. **D** Zeta potential of TIP membranes. **E** $S$ parameter of TIP25 and TIP100 membranes as characterized by the DBES characterizations. **F** Rejection of four neutral molecules (i.e., ethanol, ethylene glycol, glycerol, and glucose) with different molecular weights by TIP25 and TIP100 membranes. The error bars represent the standard deviation of the results obtained from at least three independent measurements.

nanovoid regions, as evidenced by the even deposition of gold nanotracers (Supplementary Note 18). Meanwhile, the nanovoid structures provided TIP100 with a larger effective filtration area (Fig. 3G) than that of TIP25, thereby reducing the average localized flux. Since membrane fouling has a critical dependence on water flux[47–49], the lower and more uniform local flux of TIP100 helps to mitigate membrane fouling[43]. In the current study, these effects appear to dominate over the effects of roughness and hydrophobicity. Furthermore, TIP100 exhibited high fouling reversibility with a lower irreversible flux reduction of 1.6% compared to 9.2% for TIP25. This result could be ascribed to the reduced compaction of the foulant layer for TIP100 membrane due to its lower average localized flux and more uniform flux distribution[49,50].

**Molecular insights of thermal effects on membrane formation**
To further elucidate the mechanisms of thermal-facilitated monomer diffusion, we performed molecular dynamics (MD) simulation to reveal monomer transport at molecular level. According to the simulation results, a high temperature could accelerate MPD molecules moving to the interface (Supplementary Fig. 24). Consequently, more MPD molecules accumulated at the interface (Fig. 5A, B). Such interfacial accumulation led to an increased MPD concentration gradient across the aqueous/organic interface that could facilitate its further diffusion into the organic phase (Fig. 2A). Subsequently, the molecular motion of diffused MPD and TMC were further accelerated in the organic phase with high temperature (i.e., 100 °C). In specific, MPD and TMC showed a self-diffusion coefficient of $3.8 \times 10^{-9}$ m$^2$ s$^{-1}$ and $3.3 \times 10^{-9}$ m$^2$ s$^{-1}$ at 100 °C, which is near 3 times and 6 times of the values at 25 °C, respectively (Fig. 5C). This accelerated molecular motion could increase the collision between MPD and TMC

molecules, thus enhancing their reaction efficiency. Accordingly, the reaction rate constant between MPD and TMC at 100 °C was over an order of magnitude higher than that at 25 °C (i.e., 0.99 s$^{-1}$ vs. 0.05 s$^{-1}$, Fig. 5D). The faster IP reaction would be favorable for forming a more crosslinked polyamide rejection layer with extensive nanostructures. The above enhancement on molecular diffusion and motion at a high temperature intensified the IP reaction, thereby forming a polyamide layer with a rough surface (Fig. 3A, B, G), extensive interior nanovoids (Fig. 3C, F), a highly crosslinked network (Fig. 2B), and narrowed pore size (Fig. 2F and Supplementary Fig. 11). As a result, these features allow the membrane to achieve ultra-selective rejection of various toxic micropollutants (Fig. 1D), simultaneously with high water permeance (Fig. 1A).

## Discussion
In this study, we developed an ultra-selective polyamide membrane using a TIP strategy. By raising organic phase temperature, the monomer motion and reaction were boosted and thus intensifying the IP reaction to form a highly crosslinked polyamide with a narrowed pore size. It enables the resultant TIP membrane featuring with strengthened size exclusion ability to achieve high rejection of various toxic micropollutants (e.g., 90.8% for boron, 98.0% for As (III), and > 99.0% for EDCs and antibiotics) that well exceed most RO membranes in existing literature. Meanwhile, the TIP membrane had enhanced water permeance thanks to their enlarged interior nanovoids and increased effective filtration areas. Leveraging on the combination of high water permeance and micropollutant rejection, the developed TIP membrane demonstrated transcendent water-micropollutant selectivity (e.g., water-boron). The ultra-selective rejection of micropollutants by the membrane can greatly improve the quality of

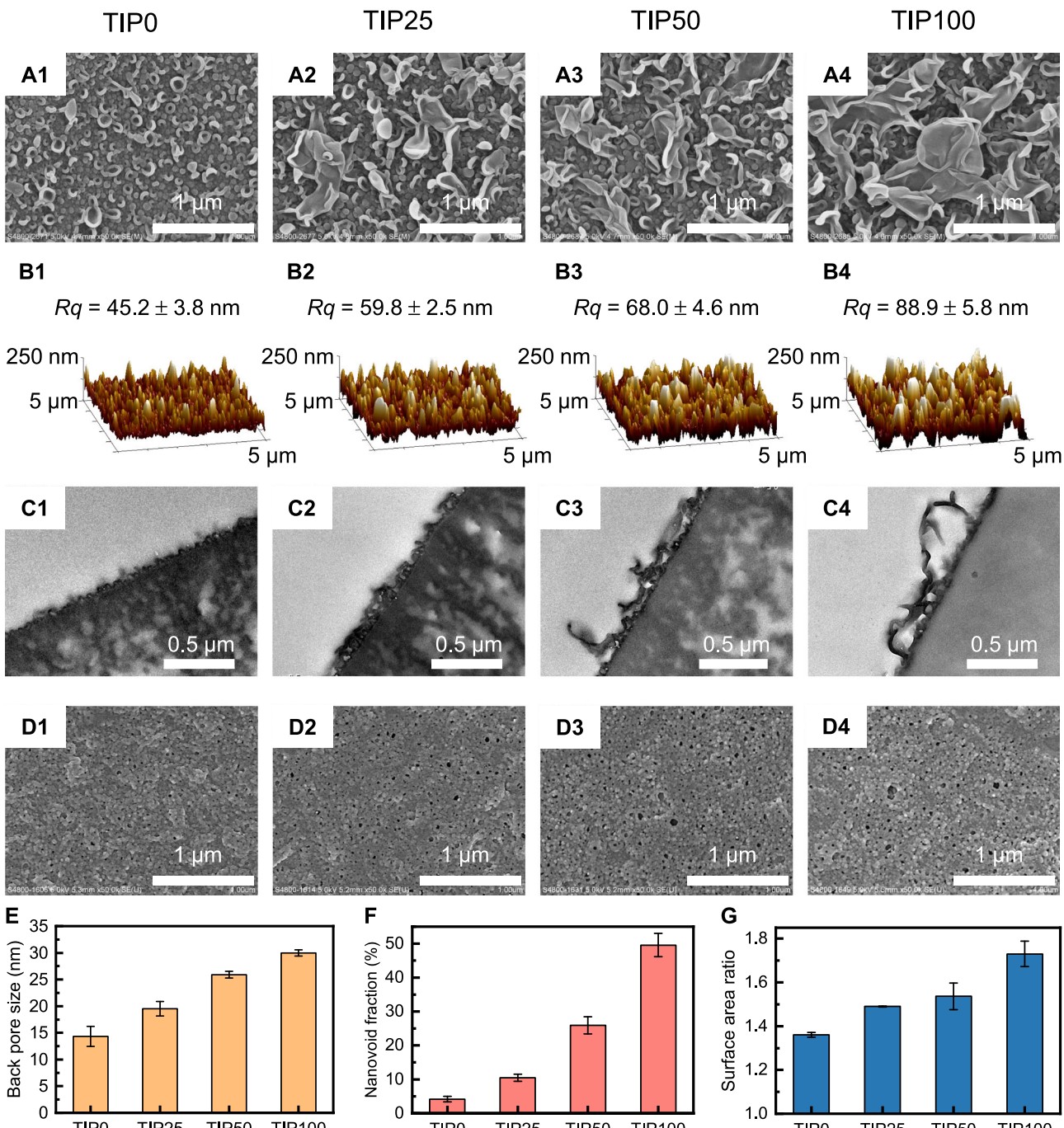

**Fig. 3 | Microscopic characterizations of various TIP RO membranes. A** Surface morphology of the membranes characterized by a scanning electron microscope (SEM). **B** Surface structures and roughness of the membranes characterized by an atomic force microscope (AFM). *Rq* is the root mean square roughness. **C** Cross-sectional structure of the membranes characterized by a TEM. **D** SEM images of back-side pores for polyamide layers. **E** Back pore size measured using the software of *Image-Pro Plus* (Supplementary Fig. 14). **F** Nanovoid fraction of the polyamide layers. This value is calculated by the area of nanovoids over the entire area of the polyamide layer based on the TEM images (Supplementary Fig. 15). **G** Surface area ratio of the membranes, which is calculated by dividing the true surface area of a membrane sample by its projected area. The error bars represent the standard deviation of the results obtained from at least three independent measurements of different membranes.

product water to ensure public safety and avoid additional treatment steps (e.g., a second-pass RO for polishing boron rejection). In addition, the better antifouling property of the TIP membrane may benefit its practical applications for wastewater treatment. Our developed TIP strategy provides an important insight into the fabrication of ultra-selective polyamide RO membranes towards highly effective removal of toxic micropollutants in membrane-based seawater, brackish water and wastewater treatment.

## Methods

### Materials and chemicals

*m*-phenylenediamine (MPD, 99%), trimesoyl chloride (TMC, 98%), and Isopar G (analytical grade) from the Sigma-Aldrich were utilized to prepare the polyamide membranes on a polysulfone (PSf) substrate (molecular weight cut-off of 67 kDa, Vontron Technology). Dimethyl-formamide (DMF, 99.5%, RCI) was used to dissolve the PSf substrate for the characterization of backside pores on polyamide layer.

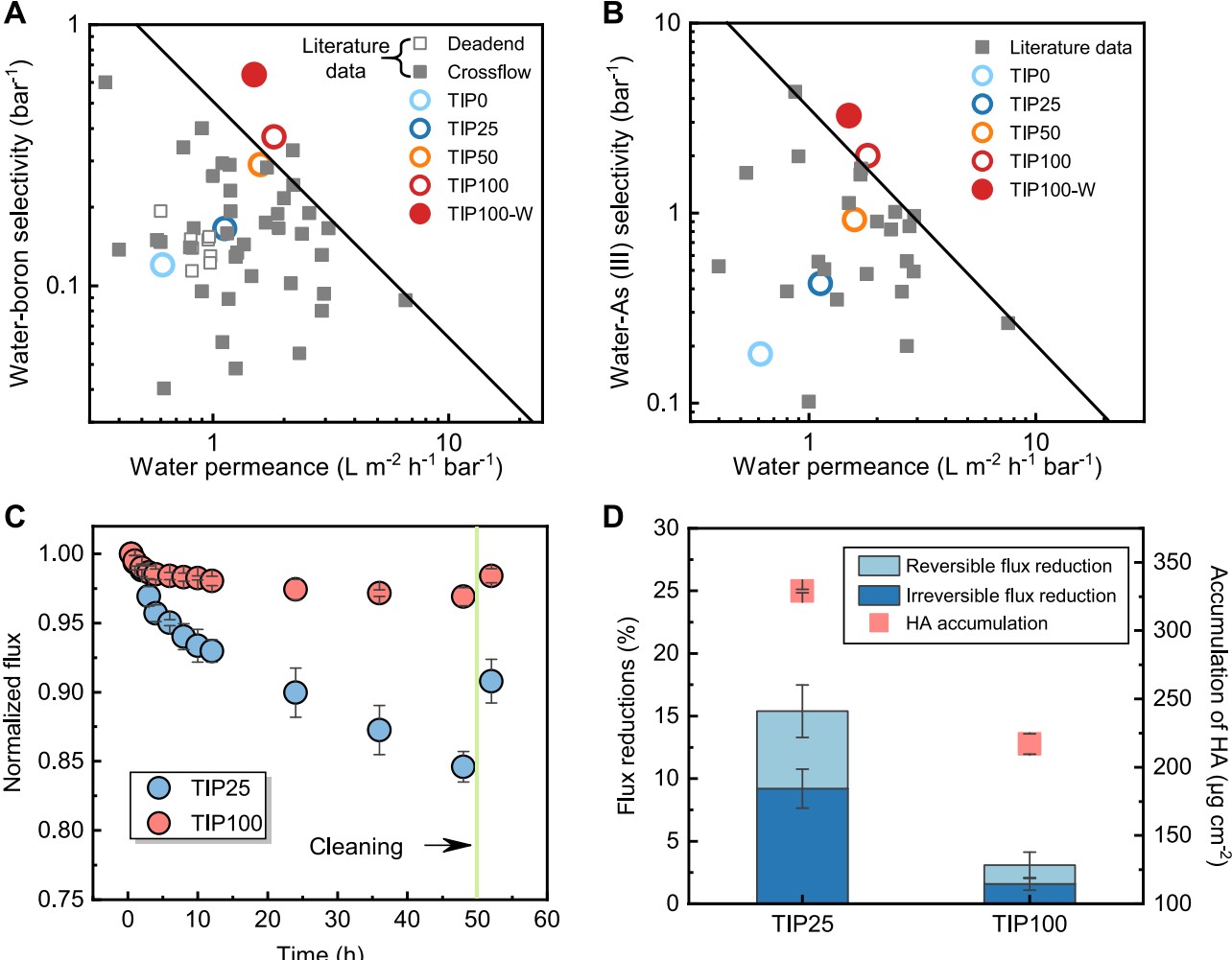

**Fig. 4 | Membrane selectivity and fouling behavior. A** Water permeance and water-boron selectivity for the TIP membranes (colored circles) compared with literature data (grey squares). All the relevant data and references have been summarized in Supplementary Table 4. **B** Water permeance and water-As (III) selectivity for the membrane prepared in this study (colored circles) compared with literature data[63] (grey squares). All data was obtained from the filtration tests under crossflow conditions at a neutral pH of 6-8. All the relevant data and references have been summarized in Supplementary Table 5. **C** Normalized flux of TIP25 and TIP100 membranes during a fouling-cleaning test over 50 h. Normalized flux is the ratio of the water flux at time $t$ ($J_t$) over the initial water flux ($J_0$, 15 L m$^{-2}$ h$^{-1}$ in this

study). The fouling test was performed using a feed solution of 0.1 g L$^{-1}$ HA and 2 g L$^{-1}$ NaCl at pH7 for 48 h. During the fouling tests, the initial flux $J_0$ for both membranes was kept the same (i.e., 15 L m$^{-2}$ h$^{-1}$) to ensure a fair comparison of fouling behaviour[64,65]. To achieve this initial flux, the applied pressure was approximately 13.6 bar for TIP25 and 8.3 bar for TIP100. The fouled membranes were cleaned with DI water, followed by the filtration of 2 g L$^{-1}$ NaCl under the same pressure for 2 h. **D** Flux reduction and HA accumulation for TIP25 and TIP100 membranes after a 48-h fouling test. The error bars represent the standard deviation obtained from at least three independent measurements of different membranes.

Hydrochloric acid (HCl, 37%, VWR) and sodium hydroxide (NaOH, analytical grade, Dieckmann) were used for pH adjustment. Ethanol (analytical grade, VWR), ethylene glycol (99%, Aladdin), glycerol (analytical grade, Leyan), and glucose (analytical grade, Uni-Chem) were used as molecular probes to evaluate the size exclusion effects of fabricated membranes. Silver nitrate (AgNO$_3$, ≥99%, Sigma-Aldrich) and nitric acid (HNO$_3$, LC-MS grade, Fisher Scientific) were used to determine the ionized carboxyl group density of polyamide layers. Sodium chloride (NaCl, analytical grade, Dieckmann), boric acid (B(OH)$_3$, analytical grade, Dieckmann), and arsenic oxide (As$_2$O$_3$, ≥99.0%, Sigma-Aldrich) were used to evaluate membrane rejection. Four EDCs (i.e., methylparaben (≥99%), ethylparaben (99%), pro-pylparaben (99%), and benzylparaben (≥99%)) and five antibiotics (i.e., sulfadiazine (≥99%), sulfamethoxazole (analytical grade), sulfa-methazine (≥99%), norfloxacin (≥98%), and ofloxacin (≥98%)) from Sigma-Aldrich were used for the rejection tests of organic micro-pollutants. Humic acid (HA, technical grade, Sigma-Aldrich) was used for membrane fouling evaluation.

## Fabrication of TIP membranes

TIP polyamide membranes were fabricated by performing the IP reaction between an aqueous solution containing 2 wt. % MPD and an organic solution of 0.1 wt. % TMC in Isopar G with temperatures of 0, 25, 50, or 100 °C on the PSf substrate (Supplementary Fig. 2). The 0 °C TMC solution was pre-cooled in a refrigerator before fur-ther usage, while the 50 °C and 100 °C TMC solutions were prepared by heating the solution to the target temperature. Briefly, the PSf substrate was first immersed in the MPD solution for 2 min and the excess solution was removed using a rubber roller. Subsequently, the TMC solution with certain temperature was poured onto the MPD-impregnated substrate to initiate the TIP reaction for 1 min. The fabricated membranes were named as TIP0, TIP25, TIP50, and TIP100, corresponding to the Isopar G temperature 0, 25, 50, and 100 °C, respectively. To further investigate the effects of post-treatment, the fabricated TIP25 and TIP100 membranes were carefully transferred into a 50 °C water bath for 10 min while the prepared membranes were denoted as TIP25-W and TIP100-W. All

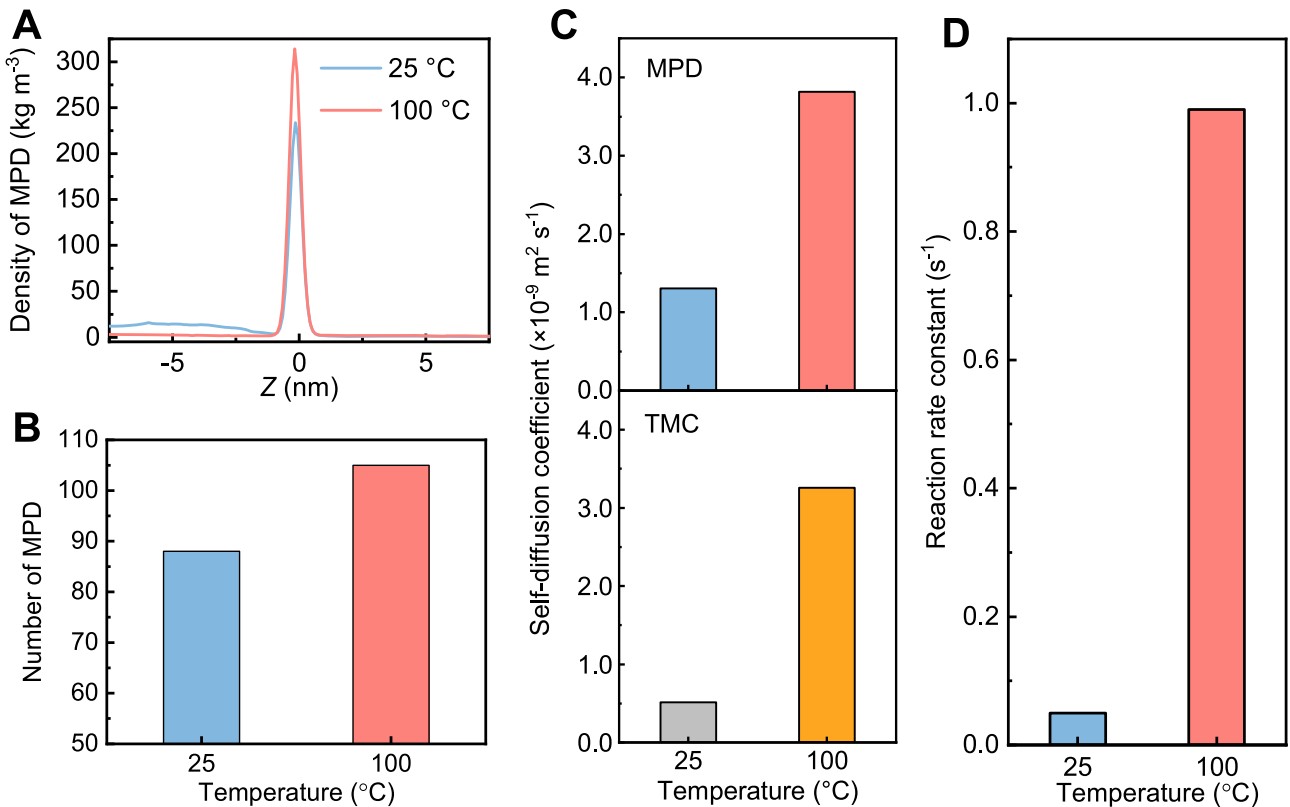

**Fig. 5 | Monomer transport and reactivity at 25 °C and 100 °C. A** Density of MPD molecules at different locations along the Z coordinate under equilibrium state. The zero point at Z coordinate represents the aqueous/organic interface while the left and right location of zero point refers to the distance away from the interface in the aqueous and organic phase, respectively. **B** Number of MPD molecules at the aqueous/organic interface under equilibrium state. The density and number of MPD were determined based on the results of molecular dynamic simulation. **C** Self-diffusion coefficient of MPD and TMC in organic phase, which was calculated using the mean square displacement (MSD) profile (Supplementary Fig. 26). **D** IP reaction rate constant calculated by the Gibbs free energy barrier of the reaction between MPD and TMC (Supplementary Fig. 27).

the fabricated membranes were thoroughly rinsed and stored with deionized (DI) water before tests.

## Characterization of TIP membranes

The surface morphology and back pores of polyamide were characterized using a SEM (S-4800, Hitachi) with an accelerating voltage of 5 kV. For back pore characterization, the PSf substrate of the membrane was first dissolved by DMF, and the separated polyamide was then transferred onto a silicon wafer with top polyamide surface facing the wafer[35]. All membranes were dried overnight in a 40 °C oven and sputter-coated with a thin gold film before the SEM characterization. The diameter of the back pores was measured from SEM images using the software of *Image-Pro Plus* (Supplementary Fig. 14). The cross-sectional morphology of a polyamide membrane was observed using a TEM (CM100, Philips). Nanovoid fraction, apparent thickness and intrinsic thickness of the polyamide membranes were determined from TEM images using the software of *Image-Pro Plus* (Supplementary Note 13 and S15). Surface roughness and surface area ratio of the polyamide membranes were assessed with an AFM (Dimension 3100, Vecco).

The functional groups of TIP polyamide membranes were analyzed using an attenuated total reflection Fourier transform infrared (ATR-FTIR) spectroscopy (Nicolet IS5, Thermo Fisher Scientific) with a wavenumber range from 600 cm$^{-1}$ to 1800 cm$^{-1}$. An X-ray photoelectron spectroscopy (XPS, Thermo Scientific K-Alpha) was used to examine the elemental composition of polyamide thereby calculating its crosslinking degree. Membrane surface charge was measured by a zeta potential analyzer (SurPASS 3, Anton Paar) at a pH range of 3–10. Membrane wetting property was assessed using a contact angle

analyzer (Attension Theta, Biolin Scientific). A Doppler Broadening Energy Spectroscope (DBES, Institute of High Energy Physics, Chinese Academy of Sciences) equipped with a $^{22}$Na source was used to determine the $S$ parameter of polyamide at a positron energy ranging from 0 to 5.2 keV, while a higher $S$ parameter indicates a larger pore size[33,34].

## Determination of ionized carboxyl group density for TIP membranes

The density of ionized carboxyl group for polyamide membranes was determined using a AgNO$_3$ titration method[31,51]. Silver ions prefer binding with ionized carboxyl groups, which originate from the hydrolysis of unreacted acyl chloride groups within the polyamide membrane. To ensure complete deprotonation of the carboxyl groups, the pH value of the AgNO$_3$ solution was adjusted to 10.5. Then, the bounded silver ions were eluted using 1 % HNO$_3$, and the concentration of leached silver ions was analyzed by an inductively coupled plasma mass spectrometry (ICP-MS, Agilent 7900). The density of ionized carboxyl group ($COO^-$) can be calculated by:

$$d(COO^-) = \frac{C_{Ag} \times V_{Ag} \times N_A}{M_{Ag} \times s_m} \tag{1}$$

where $d(COO^-)$ (nm$^{-2}$) is the density of ionized carboxyl groups, $C_{Ag}$ (g L$^{-1}$) represents silver concentration, $V_{Ag}$ (L) is the volume of elution solution, $N_A$ is the Avogadro constant (i.e., $6.02 \times 10^{23}$ mol$^{-1}$), $M_{Ag}$ (g mol$^{-1}$) is the molar weight of silver (i.e., 108 g mol$^{-1}$), and $s_m$ (m$^2$) is the surface area of the membrane coupon (i.e., $1 \times 10^{-4}$ m$^2$ in this study).

## Detection of MPD diffusion

The diffusion of MPD monomers was monitored using a ultraviolet-visible (UV) spectrophotometer (UH5300, Hitachi) at a wavelength of 290 nm (Supplementary Fig. 3)[52]. Briefly, a 0.3 mL 2 wt. % MPD aqueous solution was injected into a UV quartz cuvette. Subsequently, a 2.7 mL Isopar G solution at various temperatures (i.e., 0, 25, 50, or 100 °C, respectively) was carefully added onto the MPD solution along the inner wall of quartz cuvette using a pipette. After a 1-min interval (corresponding to the IP reaction time), the MPD absorbance signal in the organic phase was recorded.

## Evaluation of TIP membrane separation performance

The separation performance of TIP membranes was evaluated using a lab-scale crossflow filtration setup at room temperature of ~25 °C. A membrane sample was mounted in a stainless-steel filtration cell with an effective filtration area of 12 cm$^2$. Subsequently, the membrane was pre-compacted with pure water (5 L) at 17 bar for 2 h followed by the measurement of water permeance at 15.5 bar. After that, 10 g NaCl was added into the pure water to form a 2 g L$^{-1}$ NaCl solution and the NaCl rejection was tested at 15.5 bar after a 2-h filtration at 17 bar with a crossflow velocity of 22.4 cm s$^{-1}$. Water flux ($J_v$, L m$^{-2}$ h$^{-1}$), water permeance ($A$, L m$^{-2}$ h$^{-1}$ bar$^{-1}$), NaCl rejection ($R$, %), and permeability coefficient $B$ (L m$^{-2}$ h$^{-1}$) can be calculated by the following equations:

$$J_v = \frac{\Delta m}{\rho \times a \times \Delta t} \quad (2)$$

$$A = \frac{J_v}{\Delta P - \Delta \pi} \quad (3)$$

$$R = \frac{C_f - Cp}{C_f} \times 100\% \quad (4)$$

$$B = \frac{1 - R}{R} \times J_v \quad (5)$$

where $\Delta m$ (kg) is the mass of permeate water within the collecting time $\Delta t$ (h). $\rho$ (kg L$^{-1}$) is the density of permeate water, $a$ (m$^2$) is the effective filtration area, $\Delta P$ (bar) is the applied pressure, $\Delta \pi$ (bar) is the osmotic pressure difference across the tested membranes, $C_f$ and $Cp$ (g L$^{-1}$) are the concentrations of the feed and permeate solutions. The NaCl concentration was measured using a conductivity meter (Ultrameter II, Myron L).

To further evaluate the membrane separation performance for toxic micropollutants, the following feed solutions were tested: (i) 5 mg L$^{-1}$ boron in pure water (pH7), (ii) 1 mg L$^{-1}$ As (III) in pure water (pH7), and (iii) a mixture solution containing four EDCs and five antibiotics (pH7) with a concentration of 0.2 mg L$^{-1}$ for each compound in a background solution of 600 mg L$^{-1}$ NaCl. The testing conditions were identical to NaCl rejection test except a longer filtration time of 6 h for allowing the membrane to achieve stable rejection of micropollutants. The concentrations of boron and As (III) were quantified by the ICP-MS while the micropollutant concentrations were analyzed by a liquid chromatography (1260 Infinity, Agilent) with a tandem mass spectrometry (3200 QTRAP, AB SCIEX). The rejection for these contaminants can also be calculated using Eq. (4). Water-solute selectivity was defined as the ratio of $A$ to $B$.

## Evaluation of TIP membrane fouling behavior

To evaluate the fouling behavior of TIP membranes, HA was used as a model foulant. Fouling tests were also performed using the lab-scale crossflow filtration setup. Briefly, a membrane sample was pre-compacted using a 2 g L$^{-1}$ NaCl feed solution (5 L) at 17 bar with a crossflow velocity of 22.4 cm s$^{-1}$ for 12 h. Subsequently, the initial flux

($J_0$) was set to 15 L m$^{-2}$ h$^{-1}$ for both TIP25 and TIP100 membranes. To achieve this initial flux, the applied pressure was approximately 13.6 bar for TIP25 and 8.3 bar for TIP100. Then, 0.5 g HA was introduced to the NaCl feed solution to initiate the membrane fouling process. Each fouling experiment lasted for 48 h, and the measured water flux at 48 h was defined as $J_1$ (L m$^{-2}$ h$^{-1}$). The fouled membranes were cleaned using DI water under a crossflow velocity of 22.4 cm s$^{-1}$ for 0.5 h, with no pressure applied during this process. Subsequently, water flux ($J_2$, L m$^{-2}$ h$^{-1}$) was evaluated using a 2 g L$^{-1}$ NaCl feed solution under the same pressure. The irreversible flux ($R_{ir}$, %) and reversible flux ($R_r$, %) were calculated by following equations:

$$R_{ir} = \frac{J_0 - J_2}{J_0} \times 100\% \quad (6)$$

$$R_r = \frac{J_2 - J_1}{J_0} \times 100\% \quad (7)$$

In addition, HA accumulation on the membrane surface was also assessed. After a 48-h fouling test, the fouled membrane was carefully taken out from the cell and then immersed in a 0.1 M NaOH solution with moderate shaking for 12 h to extract the deposited HA[53]. The HA absorbance was tested using the UV spectrometer at a wavelength of 229 nm (Supplementary Fig. 22A), which can be further converted into concentration through the established calibration curve between UV absorbance and HA concentration (Supplementary Fig. 22B).

## Molecular dynamics simulation

The molecular dynamics (MD) simulation was conducted to elucidate the effects of thermal on monomer diffusion at molecular level using GROMACS package[54]. The velocity-Verlet algorithm was employed with a time step of 1.0 fs. All covalent bonds were constrained using the LINCS algorithm. The cut-off distance of all non-bond interactions was 1.2 nm. Long-range electrostatic interactions were computed by using the particle-mesh Ewald (PME) method, while the LJ tail correction was added to the energy and pressure. The simulation temperature was maintained by Nosé-Hoover thermostat and pressure was maintained at 1 bar by Parrinello-Rahman barostat, and the coupling time constant were set to 0.2 and 2.0 ps, respectively. Periodic boundary conditions were applied in XYZ directions.

To obtain density and number of MPD at the aqueous/organic interface, the initial configurations of simulations were generated by placing the aqueous and organic phase into the box. The lengths of XYZ directions for aqueous and organic phase are 10 nm × 10 nm × 15 nm. After reaching the equilibrated status, MPD molecules were introduced into the aqueous phase. Following previous studies[27,55,56], the OPLS all-atom force field was adopted for organic molecules (i.e., Isopar G, MPD, and TMC), while the SPCE model was used for water molecules. The simulation temperature of aqueous phase and organic phase was maintained at 298.15 K (25 °C) or 373.15 K (100 °C), respectively. The simulations were carried out in canonical ensemble and two virtual walls were added on the z-direction to keep a stable interface.

The self-diffusion coefficient of monomers was acquired through simulating the molecular motion of MPD in water and Isopar G at 298.15 K or 373.15 K, and TMC motion in organic phase at 298.15 K or 373.15 K, respectively. The simulation duration was 20 ns. Trajectories were dumped at a 0.1 ns interval and the last 5 ns simulation trajectories were adopted for further analysis. The mean square displacement (MSD) of the MPD molecules is defined by:

$$MSD = \frac{1}{N} \left\langle \sum_{i=0}^{N} |r(t) - r(0)|^2 \right\rangle \quad (8)$$

where $r(t)$ (nm) represents the position of the particle of $i$ at time $t$ (s).

The self-diffusion coefficient ($D$, m$^2$ s$^{-1}$) can be obtained from the long-time limit of the MSD using the well-known Einstein relation.

$$D = \frac{1}{6} \lim_{t \to \infty} \frac{d}{dt} \left\langle |r(t) - r(0)|^2 \right\rangle \qquad (9)$$

### Density functional theory (DFT) calculation of TIP reaction

All DFT calculations were carried out with the Gaussian 16 software at the M06-2×[57]-D3[58]/def2-SVP[59] level of theory. Gibbs free energy calculations for the equilibriums of the reactant, transition structure (searched by the Berny algorithm), and the product were performed using frequency calculations to derive the enthalpy and entropy values. The Solvation Model based on Density (SMD) model[60] was employed in this study. As this model is implicit, a dielectric constant of 2.006 was used for the solvent of Isopar G (based on previous experimental work[61]. The solvation model based on the charge density of a solute molecule[60] was employed to determine solvent effect. Reaction rate constant $u$ (s$^{-1}$) can be calculated following the Eyring equation:

$$u = \frac{k_B T}{h} \exp\left(\frac{-\Delta G}{RT}\right) \qquad (10)$$

where $k_B$ is Boltzmann constant (i.e., $1.38 \times 10^{-23}$ J K$^{-1}$), $T$ is the reaction temperature (i.e., 298.15 K and 373.15 K in this study), $h$ is the Planck constant ($6.63 \times 10^{-34}$ J s), $\Delta G$ (kJ mol$^{-1}$) is the Gibbs free energy barrier between reactants and transition structure, and $R$ is ideal gas constant (i.e., 8.31 J mol$^{-1}$ K$^{-1}$).

### Data availability

The data generated in this study are provided in the article, Supplementary Information, and Source data file. All data are available from the corresponding author upon request. Source data are provided with this paper.

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

## Acknowledgements

The work was substantially supported by grants from the Research Grants Council of the Hong Kong Special Administration Region, China (SRFS2021-7S04, GRF 17206122, and GRF 17205724 to C.Y.T.). This work was also partially supported by the Seed Funding for Strategic Interdisciplinary Research Scheme at The University of Hong Kong (102010174 to C.Y.T.). H.G. thanks the funding support from the National Natural Science Foundation of China (52300055), Natural Science Fund of Shenzhen (JCYJ20230807111702006), and Department of Education of Guangdong Province (2023KCXTD053). We appreciate the help from the Electron Microscopy Unit at The University of Hong Kong for SEM and TEM characterization.

## Author contributions

S.Z., H.G. and C.Y.T. conceived the idea and designed the research. S.Z. and W.L. conducted experiments. S.Z., L.E.P., H.G. and C.Y.T. wrote the manuscript. All authors discussed the results.

## Competing interests

The authors declare no competing interests.
