## [Peer Review file · Nature Communications]

Thermal-intensified interfacial polymerization enables ultra-selective reverse osmosis membrane for toxic micropollutant removal

Corresponding Author: Professor Chuyang Tang

Version 0:

Reviewer comments:

Reviewer #1

(Remarks to the Author)

The submitted manuscript reports the preparation of interfacially polymerized polyamide reverse osmosis membranes in the standard approach but introducing one noteworthy change which is increasing the temperature of one of the two reacting solutions (i.e., the organic solvent with the TMC monomer). The authors prepared membranes at different temperatures of the organic solvent, and performed a technically rigorous and thorough characterization of the resulting membranes in terms of their physical, chemical, and performance properties. In addition to discussing the corresponding results, the authors link the performance differences/enhancements to the known mechanisms of membrane formation and polyamide layer structure. Independent of whether or not their best membrane overcomes the permeance-selectivity tradeoff, the evaluation of the effect of temperature on membrane properties and performance is valuable. I believe this is valuable work to share with the membrane/water treatment community. I do not have major points of concern but recommend addressing the comments below to improve clarity and technical accuracy.

Main

p11: the statement that the TIP100(-W) membranes overcame the permeance-selectivity tradeoff does not seem strong for As(III) for which the TIP100(-W) membranes performed relatively similarly to the trade-off limit. In my opinion, the difference shown in Fig 4B is just not big enough to make a categorical statement about overcoming the permeance-selectivity tradeoff. As the authors may be aware, relatively small differences between experimental procedures and data analysis across studies may result in the small differences seen in Fig 4B between the trade-off line and the TIP100(-W) membranes. As an example, one factor that affects results is whether experiments are performed in dead-end or crossflow systems; dead-end experiments give lower (sometimes substantially lower) rejections (lower selectivity) than crossflow experiments. I am uncertain of whether the data in gray is all from cross flow tests or both types. I think it would be fair to say that TIP100(-W) performed at least as well as the best membranes reported in the literature (data in gray).

p7, L145-147: at this stage in the paper it is unclear what this absorbance is. It may be useful to state that this measurement was not taken from the actual interfacial polymerization reaction solutions but a separate control test evaluating MPD diffusion across the water-IsoparG interface.

p7, L145-147 and p19, L369-375: Could the authors provide evidence that the UV absorbance spectrum in isopar G does not depend on temperature? If it does, how was this taken into account in the data analysis?

p7, Fig 2A: Fig 2A seems to show UV absorbance due to MPD in Isopar G. However, the UV spectrum used by the authors as reference appears to be that of MPD in water (Supplementary Fig 2). Could the authors clarify if the spectrum in the Supplementary Fig 2 is actually in isopar G. If it is not, why not if that is the one followed in Fig 2A?

p7, Fig 2F and supplementary Fig 8: please specify how the authors obtained supplementary Fig 8 from Fig 2F.

p11, L207, and p19, section starting in L377: was the initial flux the same for both membranes? were the tests for different membranes performed at different pressures to make sure the initial fluxes matched? Please clarify.

Other

p2, L34: permeance (typo)

p4, L87: I would not use the word 'facile' here. At lab-scale, it is indeed easy to heat up the organic solvent used in the procedure. At manufacturing scale, this is a completely different challenge, the solution of which is not facile.

Reviewer #2

(Remarks to the Author)

This work reports a thermally intensified interfacial polymerization (IP) for preparing polyamide RO membranes with reduced pore sizes that enhance rejection performance, particularly for neutral contaminants that are typically difficult to remove using conventional RO membranes. Using high-temperature organic solvents during IP is a novel and facile method for tuning membrane pore structure, morphology, and properties, representing an important advancement in membrane fabrication techniques. The article is scientifically sound, and the discussions are well supported by experimental data, advanced structure characterization (such as high-resolution TEM and Doppler Broadening Energy Spectroscopy), molecular dynamics simulation, and density functional theory calculation. This article can be considered for publication in Nature Communications after addressing the following questions and comments.

1. How does a higher Isopar G temperature affect the polymer substrate and water evaporation? Did the substrate experience mechanical deformation upon exposure to the high-temperature solvent? Additionally, could the high temperature, especially 100 °C, accelerate water evaporation from the substrate, and how might this affect the membrane structure and morphology?
2. The authors are encouraged to elaborate on the structural information that can be drawn from the back pore size. Does the presence of these back pores indicate that the polyamide layer is discontinuous or contains large open pores after removing the polysulfone layer?
3. It appears that a fixed surface area ($1 \times 10^{-4} \text{ m}^2$) was used to calculate the density of ionized carboxyl groups. However, since the actual surface area of the polyamide layer increases with higher Isopar G temperature, does this imply that the actual carboxyl group density for the TIP100 membrane is even lower than the calculated value? The authors are encouraged to comment on this point.
4. The authors attribute the improved fouling resistance of the TIP100 membrane to more uniform water transport. It is worth noting that this membrane also exhibits a higher contact angle (lower hydrophilicity) and greater surface roughness, both of which are generally associated with increased fouling potential. A discussion on why fouling was not exacerbated in this case would be valuable.
5. Would the faster reaction kinetics at elevated solvent temperatures lead to less uniform distribution of pore sizes? Fig. S8 appears to suggest a broader pore size distribution.
6. How do the nanovoids remain structurally stable during the high-pressure RO process? Could the applied hydraulic pressure cause the nanovoids to compress or even collapse into the support pores?
7. What was the composition of the feed water solution used to evaluate the rejection of boron, arsenic, EDCs, and antibiotics? Was it pure deionized water or a water matrix that contained background salts or ions?

Reviewer #3

(Remarks to the Author)

The authors report that thermally intensified interfacial polymerization (TIP) enhances membrane performance by simultaneously increasing crosslinking density and introducing nanovoids. This dual effect leads to reduced pore size and improved selectivity on one hand, while also creating larger filtration areas and optimized water transport pathways on the other, thus enhancing water permeance. As a result, this strategy shows promise in mitigating the conventional tradeoff between selectivity and permeance. The topic is of broad interest and has the potential to appeal to a wide readership. However, before the manuscript is considered for publication in Nature Communications, the following concerns, particularly regarding the molecular simulations, must be addressed:

1. Clarification of post-treatment effects (Figure 1a): The post-treatment effect appears more pronounced in TIP100 compared to TIP25. Specifically, while the water permeance of TIP25-W slightly decreases relative to TIP25 (within the error bar), the reduction in TIP100-W is significantly greater. The authors should provide an explanation for this trend.
2. Surface morphology (Figure 3ab): As temperature increases, one would expect the interfacial reaction to become faster and more homogeneous, leading intuitively to a smoother surface. If the observed leaf-like features are due to byproduct acids or CO₂ degassing (nanobubbles), further characterization or mechanistic justification should be provided.
3. Definition and clarification of nanovoids (Figure 3c): The term "nanovoid" is not clearly defined in the main text. Consider integrating the explanation currently in Figure S9 directly into Figure 3c to improve clarity and consistency.
4. Literature reference (Figure 4a caption): Please provide a citation for the literature data used in Figure 4a to support comparison.
5. Figure 5e is not useful and does not provide sufficient molecular insights.

6. MSD and diffusivity (Figure 5c): For reliable self-diffusivity calculations, the MSD should be plotted on a log-log scale to confirm that the system has reached the diffusive regime (i.e., $\text{MSD} \propto t$). Although the self-diffusivities increase with temperature is expected, the current MSD plots (Figures S18 and S20) do not convincingly demonstrate the reliability of self-diffusivity calculations.

7. Missing force field details (Methodology – molecular dynamics): The manuscript does not specify the force field used for molecular dynamics simulations. The authors should cite appropriate literature validating the chosen force field or perform basic validation (e.g., against DFT data) to establish its reliability.

8. Incomplete DFT methodology (Methodology – IP reaction): The details of the DFT calculations are insufficient. Please specify the solvent model employed (e.g., implicit or explicit), the dielectric constant (if implicit), and the transition state (TS) search method used for free energy landscape calculations.

9. General comment on molecular simulations: While the molecular simulations are not the core of the study, they are intended to support and rationalize the experimental findings. However, in their current form, the simulations lack critical technical detail and rigorous analysis. Addressing points 5-8 is essential to ensure the scientific soundness and credibility of the simulation results.

Version 1:

Reviewer comments:

Reviewer #1

(Remarks to the Author)

The authors have addressed my comments effectively. I find the paper now ready for publication.

Reviewer #2

(Remarks to the Author)

The authors have adequately addressed the reviewer's comments.

Reviewer #3

(Remarks to the Author)

The authors have thoroughly addressed all of my comments. I recommend this work for publication.

The changes and revisions based on the comments from Reviewer #1

Reviewer #1 (Remarks to the Author):

The submitted manuscript reports the preparation of interfacially polymerized polyamide reverse osmosis membranes in the standard approach but introducing one noteworthy change which is increasing the temperature of one of the two reacting solutions (i.e., the organic solvent with the TMC monomer). The authors prepared membranes at different temperatures of the organic solvent, and performed a technically rigorous and thorough characterization of the resulting membranes in terms of their physical, chemical, and performance properties. In addition to discussing the corresponding results, the authors link the performance differences/enhancements to the known mechanisms of membrane formation and polyamide layer structure. Independent of whether or not their best membrane overcomes the permeance-selectivity tradeoff, the evaluation of the effect of temperature on membrane properties and performance is valuable. I believe this is valuable work to share with the membrane/water treatment community. I do not have major points of concern but recommend addressing the comments below to improve clarity and technical accuracy.

Our response:

We appreciate the reviewer's positive and constructive comments. The specific issues raised by the reviewer are addressed in the following section.

Main

1. p11: the statement that the TIP100(-W) membranes overcame the permeance-selectivity tradeoff does not seem strong for As(III) for which the TIP100(-W) membranes performed relatively similarly to the trade-off limit. In my opinion, the difference shown in Fig 4B is just not big enough to make a categorical statement about overcoming the permeance-selectivity tradeoff. As the authors may be aware, relatively small differences between experimental procedures and data analysis across studies may result in the small differences seen in Fig 4B between the trade-off line and the TIP100(-W) membranes. As an example, one factor that affects results is whether experiments are performed in dead-end or crossflow systems; dead-end experiments give lower (sometimes substantially lower) rejections (lower selectivity) than crossflow experiments. I am uncertain of whether the data in gray is all from cross flow tests or both types. I think it would be fair to say that TIP100(-W) performed at least as well as the best membranes reported in the literature (data in gray).

Our response:

We thank the reviewer's suggestion. Indeed, filtration mode (e.g., dead-end or crossflow) would have important effects on the experimental results. We would like to clarify that all datapoints in Fig. 4B were acquired from the tests using the crossflow filtration system. To address the comment, we have revised the caption of Fig. 4B as:

Line 224-Line 227:

Fig. 4| (B) Water permeance and water-As (III) selectivity for the membrane prepared in this study (colored squares) compared with literature data (grey squares). All data was obtained from the filtration tests under crossflow conditions at a neutral pH of 6-8.

In addition, we have revised the discussion about water permeance and water-As (III) selectivity of our TIP100(-W) membrane in **Line 240-Line 243:**

A high organic phase temperature greatly enhanced membrane selectivity. Specifically, the TIP100 membrane demonstrated ultrahigh water-boron selectivity (Fig. 4A) that overperforms the reported data in existing literature. At the same time, this membrane performs at least as well as the best membranes reported in the literature for water-As(III) selectivity (Fig. 4B).

2. p7, L145-147: at this stage in the paper it is unclear what this absorbance is. It may be useful to state that this measurement was not taken from the actual interfacial polymerization reaction solutions but a separate control test evaluating MPD diffusion across the water-IsoparG interface.

Our response:

Following the reviewer's suggestion, we have included an explanation about MPD diffusion measurement in the revised manuscript.

Line 137-Line 141:

Fig. 2| Thermal effects on MPD diffusion and membrane properties. (A) Ultraviolet-visible (UV) absorbance of MPD diffused into Isopar G at different temperatures (i.e., 0 °C, 25 °C, 50 °C, and 100 °C). The MPD diffusion measurement was not taken from the actual IP reaction solutions but in a separate control test to evaluate MPD diffusion across the water-Isopar G interface (Supplementary Fig. 4).

In addition, the following Supplementary Fig. 4 was added for illustration purposes.

Supplementary Fig. 4| Schematic diagram of detecting MPD diffusion from water to Isopar G.

The diffusion of MPD monomers was measured using a UV spectrophotometer at a wavelength of 290 nm (Supplementary Fig. 3). Briefly, a 0.3 mL 2 wt. % MPD aqueous solution was first injected into a UV quartz cuvette (Supplementary Fig. 4). Subsequently, a 2.7 mL Isopar G solution at various temperatures (0 °C, 25 °C, 50 °C, or 100 °C, respectively) was carefully added on top of the MPD solution along the inner wall of the quartz cuvette using a pipette. After a duration of 1 min (corresponding to the IP reaction time), the MPD absorbance signal was measured for the organic phase (Isopar G).

3. p7, L145-147 and p19, L369-375: Could the authors provide evidence that the UV absorbance spectrum in isopar G does not depend on temperature? If it does, how was this taken into account in the data analysis?

Our response:

We have measured the UV absorbance spectra of MPD in Isopar G at different temperatures. The results have been included in the **Supplementary information section S3**.

S3. Ultraviolet-visible (UV) absorbance of MPD

Supplementary Fig. 3| The UV absorbance spectra of MPD in Isopar G at different temperatures with wavelengths ranging from 200 to 500 nm. The inset is the UV absorbance at 290 nm at different temperatures. The error bars represent the standard deviation of the results obtained from at least three independent measurements.

We further analyzed the UV absorbance spectra of MPD in Isopar G at different temperatures (i.e., 0 °C, 25 °C, 50 °C, and 100 °C). Briefly, 2 g MPD monomers were first added into 50 mL Isopar G at room temperature (i.e., 25 °C), followed by an ultrasound mixing for 30 min. A 3 mL of the upper solution was then taken, and its temperature was adjusted to the target temperature (0 °C, 25 °C, 50 °C, or 100 °C) through cooling or heating. Subsequently, UV absorbance spectra of MPD in Isopar G at each temperature were measured (Supplementary Fig. 3), using that of pure Isopar G as the blank. An identical absorbance peak at ~290 nm with similar intensity was observed for MPD in Isopar G at different temperatures. These results suggested that temperature has no obvious effect on the UV absorbance of MPD monomers in Isopar G.

4. p7, Fig 2A: Fig 2A seems to show UV absorbance due to MPD in Isopar G. However, the UV spectrum used by the authors as reference appears to be that of MPD in water (Supplementary Fig 2). Could the authors clarify if the spectrum in the Supplementary Fig 2 is actually in isopar G. If it is not, why not if that is the one followed in Fig 2A?

Our response:

The reviewer raised a good point. Our original UV spectrum measurements of MPD were conducted in water (Supplementary Fig. 2). We have performed additional measurements to analyze the UV absorbance of MPD in Isopar G (**Supplementary information section S4**). Following the reviewer's comment, we have replaced the original Supplementary Fig. 2 using UV absorbance of MPD in Isopar G. The related discussion has also been updated. See detailed changes in our response to comment #3.

As MPD in water and in Isopar G both show an obvious characteristic peak at ~290 nm, this revision does not affect the results or discussion in the manuscript.

5. p7, Fig 2F and supplementary Fig 8: please specify how the authors obtained supplementary Fig 8 from Fig 2F.

Our response:

The method for membrane pore size assessment using the rejection of neutral molecules was adopted from the literature.^{S2-S4} Details have been included in the **Supplementary information section S10**.

S10. Pore size distribution of TIP membranes

We evaluated the pore size distribution of TIP25 and TIP100 based on their rejection of neutral solutes with different molecular weight (Mw, g mol⁻¹), following the reported method.²⁻⁴ Briefly, rejection of ethanol (46.1 g mol⁻¹), ethylene glycol (62.1 g mol⁻¹), glycerol (92.1 g mol⁻¹), and glucose (180.2 g mol⁻¹) was evaluated (**Fig. 2F**). The rejection for each molecule was plotted against the corresponding molecular radius (r_s , nm), with r_s calculated by the following equation:⁵

$$\log r_s = -1.3363 + 0.395 \times \log_{10} Mw \quad (S1)$$

By assuming a log-normal pore size distribution, the rejection-molecular radius plot was fitted using the “log-normal CDF” function in the *Origin* software. The pore size distribution of the polyamide membrane was calculated by the probability density function:^{2,3}

$$\frac{dF(r_p)}{dr_p} = \frac{1}{r_p \rho \sqrt{2\pi}} \exp \left[-\frac{(\ln r_p - \ln \mu_p)^2}{2\sigma^2} \right] \quad (S2)$$

where r_p is the membrane pore radius (nm) and μ_p refers to the geometric mean radius (nm) of the solute at 50% rejection, while σ denotes the geometric standard deviation calculated as $(\ln \mu_p - \ln \mu_d)$, with μ_d representing the geometric mean radius (nm) of the solute corresponding to 84.13% rejection.

6. p11, L207, and p19, section starting in L377: was the initial flux the same for both membranes? were the tests for different membranes performed at different pressures to make sure the initial fluxes matched? Please clarify.

Our response:

For membrane fouling tests (original text in p11, L207), the initial flux was the same for both membranes. To further address the reviewer’s comment, we have included additional information of fouling experiments in the revised manuscript.

Line 228-Line 234:

Fig. 4 | ... (C) Normalized flux of TIP25 and TIP100 membranes during a fouling-cleaning test over 50 h. Normalized flux is the ratio of the water flux at time t (J_t) over the initial water flux (J_0). During the fouling tests, the initial flux J_0 for both membranes was kept the same (i.e., 15 L m⁻² h⁻¹) to ensure a fair comparison of fouling behaviour.^{47, 48} To achieve this initial flux, the applied pressure was approximately 13.6 bar for TIP25 and 8.3 bar for TIP100.

Line 444-Line 446:

... the initial flux (J_0) was set to $15 \text{ L m}^{-2} \text{ h}^{-1}$ for both TIP25 and TIP100 membranes. To achieve this initial flux, the applied pressure was approximately 13.6 bar for TIP25 and 8.3 bar for TIP100.

For the evaluation of membrane water permeance and solute rejection (original text in p19, L377), a fixed applied pressure of 15.5 bar was used. This information is already stated in the text (**Line 412-Line 416** in the revision).

Other

7. p2, L34: permeance (typo)

Our response:

The spelling of the word has been corrected.

8. p4, L87: *I would not use the word 'facile' here. At lab-scale, it is indeed easy to heat up the organic solvent used in the procedure. At manufacturing scale, this is a completely different challenge, the solution of which is not facile.*

Our response:

We have replaced “facile” with “effective” in the revision.

The changes and revisions based on the comments from Reviewer #2

Reviewer #2 (Remarks to the Author):

This work reports a thermally intensified interfacial polymerization (IP) for preparing polyamide RO membranes with reduced pore sizes that enhance rejection performance, particularly for neutral contaminants that are typically difficult to remove using conventional RO membranes. Using high-temperature organic solvents during IP is a novel and facile method for tuning membrane pore structure, morphology, and properties, representing an important advancement in membrane fabrication techniques. The article is scientifically sound, and the discussions are well supported by experimental data, advanced structure characterization (such as high-resolution TEM and Doppler Broadening Energy Spectroscopy), molecular dynamics simulation, and density functional theory calculation. This article can be considered for publication in Nature Communications after addressing the following questions and comments.

Our response:

We thank the reviewer for the positive and constructive comments. The specific issues raised by the reviewer are addressed in the following section.

1. How does a higher Isopar G temperature affect the polymer substrate and water evaporation? Did the substrate experience mechanical deformation upon exposure to the high-temperature solvent? Additionally, could the high temperature, especially 100 °C, accelerate water evaporation from the substrate, and how might this affect the membrane structure and morphology?

Our response:

To address the reviewer's comment, we conducted additional experiments and characterizations to assess the impacts of Isopar G temperature on the morphology, structure, and permeance of the substrate. The results indicate that the high temperature (i.e., 100 °C) did not cause mechanical deformation of the substrate (**Supplementary information section S1**).

S1. Effects of high-temperature Isopar G on the substrate

Supplementary Fig. 1 | (A) SEM surface morphology, (B) SEM cross-sectional structures,

and (C) water permeance of the pristine PSf substrate and the substrate immersed in 100 °C Isopar G for 1 min. Filtration test conditions: applied pressure of 1 bar; crossflow velocity of 22.4 cm s⁻¹; pure water as the feed solution; pre-compaction time of 2 h. The error bars represent the standard deviation of the results obtained from at least three independent measurements of different membranes.

SEM characterization does not show obvious changes for the PSf substrate after being immersed in the 100 °C Isopar G (Supplementary Fig. 1A and Fig. 1B). Moreover, the water permeance of the treated substrate ($72.0 \pm 1.7 \text{ L m}^{-2} \text{ h}^{-1} \text{ bar}^{-1}$) was comparable to that of the pristine substrate ($73.3 \pm 3.5 \text{ L m}^{-2} \text{ h}^{-1} \text{ bar}^{-1}$, Supplementary Fig. 1C). These results suggested that high-temperature Isopar G does not significantly alter the structure and properties of the substrate.

To further address the reviewer's comment on potential water evaporation under the high temperature, we performed additional experiments, and the results have been included in the revised **Supplementary information Section S11**.

S11. Evaluation of gas/vapor generation during IP reaction

Supplementary Fig. 12 (A) Schematic diagram of the custom-made setup for evaluating gas/vapor generation. This figure was adapted from Reference 7 with copyright permission. (B) Displaced water volume resulting from the addition of Isopar G (1 mL) at different temperatures (25 and 100 °C), without or with TMC (0.1 wt.%), in an MPD-prefilled (5 mL, 2 wt.%) airtight flask. The results were recorded after the system returned to the room temperature (~ 25 °C) to eliminate the effect of thermal expansion of internal gases caused by high temperature. The error bars represent the standard deviation of the results obtained from at least three independent measurements.

The following discussion was included in **Line 228-Line 232, Supplementary information**.

... It is worthwhile noting that the 100 °C Isopar G may also accelerate the water evaporation, thereby leading to more released vapor (Supplementary Fig. 12B). The greatly enhanced gas/vapor generation, upon the encapsulation by the nascent polyamide, enhances nanovoid formation.^{7, 10}

2. The authors are encouraged to elaborate on the structural information that can be drawn from the back pore size. Does the presence of these back pores indicate that the polyamide layer

is discontinuous or contains large open pores after removing the polysulfone layer?

Our response:

We appreciate the reviewer's suggestion. Additional discussion about the back pores has been included in the revised manuscript.

Line 199-Line 209:

Briefly, the increased temperature and H^+ (a byproduct of IP) could facilitate interfacial degassing of CO_2 nanobubbles from the aqueous solution ($HCO_3^- + H^+ \xrightarrow{\Delta} CO_2 \uparrow + H_2O$), which were further captured by the nascent polyamide thereby forming the extensive nanovoid structure. The additional gas generated after the formation of this nascent polyamide film will have to escape from the back side, forming the back opening structure of the polyamide (Supplementary Fig. 13 and Supplementary information section S12).^{37, 42} These back-side openings became larger at higher temperature conditions (e.g., 14.3 nm for TIP0 vs. 30.0 nm for TIP100, Fig. 3D and 3E). According to literature,⁴¹⁻⁴³ the size of the openings is well correlated to the intensity of degassing. Furthermore, the back-side openings connect the nanovoids in the polyamide to the substrate pores (Supplementary Fig. 13).^{37, 41, 42, 44}

Supplementary Fig. 13 | Schematic diagram of back-side pore formation during the IP reaction.

3. It appears that a fixed surface area ($1 \times 10^{-4} m^2$) was used to calculate the density of ionized carboxyl groups. However, since the actual surface area of the polyamide layer increases with higher Isopar G temperature, does this imply that the actual carboxyl group density for the TIP100 membrane is even lower than the calculated value? The authors are encouraged to comment on this point.

Our response:

In the current study, we have reported the carboxyl group density based on the projected membrane area, which has been commonly adopted by other groups.^{R1, R2} To allow better comparison with literature data, we have decided to keep these original results.

The reviewer has a good point on the potential influence of increased effective surface area of the polyamide layer on carboxyl group density. Accordingly, we also added carboxyl group density normalized by the actual surface area (Supplementary Fig. 8) and additional discussion has been included in the revision.

Line 156-Line 163

Consistently, the density of ionized carboxyl groups (formed by hydrolysis of unreacted acyl chloride groups) decreased from 38.2 nm^{-2} for TIP0 to 12.5 nm^{-2} for TIP100 (Fig. 2C), resulting in a less negatively charged membrane surface (Fig. 2D). It is worthwhile noting that these carboxyl group densities were estimated based on the projected membrane area, as commonly practiced in the literature.^{32, 34} However, since the actual surface area of the polyamide layer increases with higher Isopar G temperature, the carboxyl group density normalized by the actual surface area is even lower (28.1 nm^{-2} for TIP0 vs. 7.2 nm^{-2} for TIP100, Supplementary Fig. 8).

Supplementary Fig. 8 Carboxyl group density of TIP membranes normalized by the actual membrane surface area. The error bars represent the standard deviation of the results obtained from at least three independent measurements of different membranes.

References:

R1. Chen, D., Werber, J.R., Zhao, X. & Elimelech, M. A facile method to quantify the carboxyl group areal density in the active layer of polyamide thin-film composite membranes. *J. Membr. Sci.* **534**, 100-108 (2017).

R2. Yao, Y. et al. More resilient polyester membranes for high-performance reverse osmosis desalination. *Science* **384**, 333-338 (2024).

4. The authors attribute the improved fouling resistance of the TIP100 membrane to more uniform water transport. It is worth noting that this membrane also exhibits a higher contact angle (lower hydrophilicity) and greater surface roughness, both of which are generally associated with increased fouling potential. A discussion on why fouling was not exacerbated in this case would be valuable.

Our response:

We have included more discussions about membrane fouling resistance of the TIP100 membrane in the revised manuscript.

Line 251-Line 262:

We further evaluated the fouling behavior of TIP25 and TIP100 membranes using humic acid (HA) as the foulant. TIP100 membrane demonstrated superior antifouling performance (Fig. 4C) and less HA accumulation on its surface (Fig. 4D), despite its

greater roughness (**Fig. 3B**) and hydrophobicity (Supplementary Fig. 9). The reduced fouling propensity can be attributed to the extensive nanovoid structures of the membrane.⁴⁹⁻⁵¹ Such structures could allow more uniform water transport and flux distribution near the extensive nanovoid regions, as evidenced by the even deposition of gold nano-tracers (Supplementary information section S18). Meanwhile, the nanovoid structures provided TIP100 with a larger effective filtration area (**Fig. 3G**) than that of TIP25, thereby reducing the average localized flux. Since membrane fouling has a critical dependence on water flux,⁵²⁻⁵⁴ the lower and more uniform local flux of TIP100 helps to mitigate membrane fouling.⁴⁵ In the current study, these effects appear to dominate over the effects of roughness and hydrophobicity.

5. *Would the faster reaction kinetics at elevated solvent temperatures lead to less uniform distribution of pore sizes? Fig. S8 appears to suggest a broader pore size distribution.*

Our response:

The reviewer has a good point! We have included additional discussion of the pore size distribution in the revised **Supplementary information section S10**.

... The TIP100 membrane showed a broader pore size distribution (Supplementary Fig. 11), which could be attributed to the faster reaction kinetics at elevated temperatures.⁶ Nevertheless, this membrane had a smaller average pore radius of 0.17 nm, with majority of pores smaller than those of TIP25. The smaller average pore size of TIP100, beneficial for enhancing membrane rejection, can be explained by intensified IP reaction and thus higher crosslinking degree (**Fig. 2B**) at higher temperature.

6. *How do the nanovoids remain structurally stable during the high-pressure RO process? Could the applied hydraulic pressure cause the nanovoids to compress or even collapse into the support pores?*

Our response:

To address the reviewer's comment on the structural stability of nanovoids under high-pressure RO process, we conducted additional experiments to test membrane performance under various applied pressures up to 55 bar. The membrane nanovoid structure was also characterized after the high-pressure filtration tests. The results have been included in the revised **Supplementary information section S14**.

S14. Evaluation of structural stability of nanovoids

Supplementary Fig. 16. (A) Pure water permeance and NaCl rejection of TIP100 membranes measured at 15.5, 35, and 55 bar, respectively. The membrane was pre-compacted with pure water at the specified pressure for 2 h, followed by the pure water flux measurement. Subsequently, 2 g L⁻¹ NaCl solution was used as the feed solution, and NaCl rejection was determined after another 2-h filtration at the same pressure. After the filtration tests, membrane samples were rinsed with pure water for further TEM characterization. (B) TEM cross-sectional morphologies of the TIP100 membrane after filtration test at 15.5 bar (B1), 35 bar (B2), and 55 bar (B3). The error bars represent the standard deviation of the results obtained from at least three independent measurements of different membranes.

To evaluate the structural stability of nanovoids within the polyamide membranes, we performed additional filtration tests for TIP100 membranes at 15.5, 35, and 55 bar, respectively (Supplementary Fig. 16A). The NaCl rejection of the membrane was well maintained. The slight decrease in water permeance can be attributed to membrane compaction at higher applied pressure. Meanwhile, the nanovoid-containing structure was still observed even after the high-pressure filtration under 55 bar (Supplementary Fig. 16B). This result is also consistent with literature.¹¹

We note that commercial membranes were also able to maintain their nanovoid structure after high-pressure testing (e.g., BW30 after humic acid fouling).^{R3}

References:

R3. Davenport, D.M. et al. Thin film composite membrane compaction in high-pressure reverse osmosis. *J. Membr. Sci.* **610**, 118268 (2020).

7. What was the composition of the feed water solution used to evaluate the rejection of boron, arsenic, EDCs, and antibiotics? Was it pure deionized water or a water matrix that contained background salts or ions?

Our response:

We have included the composition of the feed solution in the revised manuscript.

Line 428-Line 432:

To further evaluate the membrane separation performance for toxic micropollutants, the following feed solutions were tested: (i) 5 mg L⁻¹ boron in pure water (pH7), (ii) 1 mg L⁻¹

As (III) in pure water (pH7), and (iii) a mixture solution containing four EDCs and five antibiotics (pH7) with a concentration of 0.2 mg L^{-1} for each compound in a background solution of 600 mg L^{-1} NaCl.

The changes and revisions based on the comments from Reviewer #3

Reviewer #3 (Remarks to the Author):

The authors report that thermally intensified interfacial polymerization (TIP) enhances membrane performance by simultaneously increasing crosslinking density and introducing nanovoids. This dual effect leads to reduced pore size and improved selectivity on one hand, while also creating larger filtration areas and optimized water transport pathways on the other, thus enhancing water permeance. As a result, this strategy shows promise in mitigating the conventional tradeoff between selectivity and permeance. The topic is of broad interest and has the potential to appeal to a wide readership. However, before the manuscript is considered for publication in Nature Communications, the following concerns, particularly regarding the molecular simulations, must be addressed:

Our response:

We appreciate the reviewer's positive and constructive comments. The specific issues raised by the reviewer are addressed in the following section.

1. Clarification of post-treatment effects (Figure 1a): The post-treatment effect appears more pronounced in TIP100 compared to TIP25. Specifically, while the water permeance of TIP25-W slightly decreases relative to TIP25 (within the error bar), the reduction in TIP100-W is significantly greater. The authors should provide an explanation for this trend.

Our response:

To address the reviewer's comments, we have included additional discussion in the revised manuscript.

Line 114-Line124:

In this regard, we prepared two post-treated polyamide membranes (TIP25-W and TIP100-W, with W indicating heat treatment in water), and both membranes showed higher NaCl rejection compared to their respective counterparts without post-heating (**Fig. 1B**). Whereas TIP25-W exhibited comparable water permeance with TIP25, TIP100-W showed a 17.6% reduction in water permeance compared with TIP100 (**Fig. 1A**). For TIP25-W that is post-treated at relatively low temperature, its crosslinking degree remained relatively low (Supplementary Fig. 10). In contrast, TIP100-W reached a very high crosslinking degree of 89.1% after the heat treatment, which partially explains moderate loss of water permeance. Furthermore, the surface of TIP100-W became significantly more hydrophobic compared to TIP100 (Supplementary Fig. 9), which may further increase the resistance to water transport.

2. Surface morphology (Figure 3ab): As temperature increases, one would expect the interfacial reaction to become faster and more homogeneous, leading intuitively to a smoother surface. If the observed leaf-like features are due to byproduct acids or CO₂ degassing (nanobubbles), further characterization or mechanistic justification should be provided.

Our response:

The mechanisms and evidence of nanofoaming during IP reaction have been comprehensively summarized in a recent review paper (Reference 42 in the revised manuscript).

To further address the reviewer's comment, we conducted additional experiments to assess the interfacial degassing during the IP reaction. The results have been included in the revised **Supplementary information Section S11**.

S11. Evaluation of gas/vapor generation during IP reaction

Supplementary Fig. 12 (A) Schematic diagram of the custom-made setup for evaluating gas/vapor generation. The airtight flask is pre-filled with 5 mL MPD (2 wt.%) and connected to a water column for the measurement of gas/vapor production. This figure was modified from Reference 7 with copyright permission. (B) Displaced water volume resulting from the addition of Isopar G (1 mL) at different temperatures (25 and 100 °C), without or with TMC (0.1 wt.%). The results were recorded after the system returned to the room temperature (~ 25 °C) to eliminate the effect of thermal expansion of internal gases caused by high temperature. The error bars represent the standard deviation of the results obtained from at least three independent measurements.

To further verify the interfacial degassing, we measured the volume of released gas through the displacement of water column in a custom-designed device (Supplementary Fig. 12A).⁷ Compared to the addition of Isopar G only, the addition of Isopar G with TMC resulted in much greater displacement (Supplementary Fig. 12B), which demonstrates the interfacial degassing during the IP reaction. Moreover, the Isopar G/TMC solution with a higher temperature led to an increased displaced volume, which can be explained by the promoted degassing/vaporization due to the intensified IP reaction^{8,9} and the reduced gas solubility at the higher temperature.

3. *Definition and clarification of nanovoids (Figure 3c): The term “nanovoid” is not clearly defined in the main text. Consider integrating the explanation currently in Figure S9 directly into Figure 3c to improve clarity and consistency.*

Our response:

Following the reviewer's suggestion, we have added the definition of “nanovoid” in the revised manuscript.

Line 192-Line 196:

In addition, TEM characterizations revealed the presence of more prominent nanovoids within the polyamide rejection layer formed at higher temperature (Fig. 3C). These nanovoids, formed by the encapsulation of degassed nanobubbles by the nascent polyamide during IP reaction,^{37, 38} are enclosed between the polyamide film and the substrate (as illustrated by the red-highlighted area, Supplementary Fig. 15B)

4. Literature reference (Figure 4a caption): Please provide a citation for the literature data used in Figure 4a to support comparison.

Our response:

To address the reviewer's concern, all the relevant data and references have been summarized and included in the Supplementary Table 4.

5. Figure 5e is not useful and does not provide sufficient molecular insights.

Our response:

We thank the reviewer's suggestion. Fig. 5E has been deleted in the revised manuscript. This change does not affect the overall structure and discussion of the manuscript.

6. MSD and diffusivity (Figure 5c): For reliable self-diffusivity calculations, the MSD should be plotted on a log-log scale to confirm that the system has reached the diffusive regime (i.e., $MSD \propto t$). Although the self-diffusivities increase with temperature is expected, the current MSD plots (Figures S18 and S20) do not convincingly demonstrate the reliability of self-diffusivity calculations.

Our response:

Following the reviewer's suggestions, we have plotted the MSD profile on a log-log scale and updated the results in the revised **Supplementary information**. After the revision, the overall trend is not changed.

Line 365-Line 367:

Supplementary Fig. 24 (A) Mean squared displacement profile versus diffusion time plotted on a log-log scale, and (B) self-diffusion coefficient of MPD molecules in water at 25 °C and 100 °C, respectively.

Line 373-Line 375:

... As shown in Supplementary Fig. 24, the log-log plot of mean squared displacement (MSD) versus diffusion time reveals a linear relationship ($\text{MSD} \propto t$), confirming that the system has reached the diffusive regime.

Line 390-Line391

Supplementary Fig. 26 Mean squared displacement profile versus diffusion time plotted on a log-log scale for (A) MPD and (B) TMC monomers in the organic phase of Isopar G at 25 °C and 100 °C, respectively.

7. *Missing force field details (Methodology – molecular dynamics): The manuscript does not specify the force field used for molecular dynamics simulations. The authors should cite appropriate literature validating the chosen force field or perform basic validation (e.g., against DFT data) to establish its reliability.*

Our response:

We have included more detailed information about the molecular dynamics calculation into the revised manuscript to address the reviewer's comment.

Line 476-Line 478:

... Following previous studies,^{27,59,60} the OPLS all-atom force field was adopted for organic molecules (i.e., Isopar G, MPD, and TMC), while the SPCE model was used for water molecules.

References:

27. Shen, L. et al. Polyamide-based membranes with structural homogeneity for ultrafast molecular sieving. *Nat Commun.* **13**, 500 (2022).

59. Zhao, C. et al. Polyamide membranes with nanoscale ordered structures for fast permeation and highly selective ion-ion separation. *Nat. Commun.* **14**, 1112 (2023).

60. Zhang, Y. et al. Ice-confined synthesis of highly ionized 3D-quasilayered polyamide nanofiltration membranes. *Science* **382**, 202-206 (2023).

8. *Incomplete DFT methodology (Methodology – IP reaction): The details of the DFT calculations are insufficient. Please specify the solvent model employed (e.g., implicit or explicit), the dielectric constant (if implicit), and the transition state (TS) search method used for free energy landscape calculations.*

Our response:

We have included the details of the DFT calculations in the revised manuscript.

Line 496-Line 500:

All calculations were carried out using Gaussian 16 software at the M06-2X⁶¹/D3⁶²/def2-SVP⁶³ level of theory. Gibbs free energy calculations for the equilibria of the reactant, transition structure (searched by the Berny algorithm), and the product were performed using frequency calculations to obtain the enthalpy and entropy values. The Solvation Model based on Density (SMD) model⁶⁴ was employed in this study. As this model is implicit, a dielectric constant of 2.006 was used for the solvent of Isopar G (based on previous experimental work⁶⁵).

9. General comment on molecular simulations: While the molecular simulations are not the core of the study, they are intended to support and rationalize the experimental findings. However, in their current form, the simulations lack critical technical detail and rigorous analysis. Addressing points 5-8 is essential to ensure the scientific soundness and credibility of the simulation results.

Our response:

Thanks for the reviewer's constructive comments. Please refer to our detailed reply to Comments #5-#8.

Response to reviewers' comments

Reviewer #1 (Remarks to the Author):

The authors have addressed my comments effectively. I find the paper now ready for publication.

Reviewer #2 (Remarks to the Author):

The authors have adequately addressed the reviewer's comments.

Reviewer #3 (Remarks to the Author):

The authors have thoroughly addressed all of my comments. I recommend this work for publication.

Our response:

We thank all the reviewers for their recommendations for publication.